biomedical engineering/biomechanics/physiology

cerebral palsy, deep learning, feature tracking, pose estimation, SATCo, video analysis

**Author for correspondence:**
Ryan Cunningham
e-mail: ryan.cunningham@mmu.ac.uk

# Fully automated image-based estimation of postural point-features in children with cerebral palsy using deep learning

Ryan Cunningham[1,2], María B. Sánchez[1,3], Penelope B. Butler[1], Matthew J. Southgate[1] and Ian D. Loram[1]

[1]Research Centre for Musculoskeletal Science & Sports Medicine, [2]Centre for Advanced Computational Science, and [3]Department of Health Professions, Manchester Metropolitan University, Manchester, UK

RC, 0000-0001-6883-6515; MBS, 0000-0002-4099-3970; MJS, 0000-0001-7350-2448; IDL, 0000-0001-8125-6320

The aim of this study was to provide automated identification of postural point-features required to estimate the location and orientation of the head, multi-segmented trunk and arms from videos of the clinical test 'Segmental Assessment of Trunk Control' (SATCo). Three expert operators manually annotated 13 point-features in every fourth image of 177 short (5–10 s) videos (25 Hz) of 12 children with cerebral palsy (aged: 4.52 ± 2.4 years), participating in SATCo testing. Linear interpolation for the remaining images resulted in 30 825 annotated images. Convolutional neural networks were trained with cross-validation, giving held-out test results for all children. The point-features were estimated with error 4.4 ± 3.8 pixels at approximately 100 images per second. Truncal segment angles (head, neck and six thoraco-lumbar–pelvic segments) were estimated with error 6.4 ± 2.8°, allowing accurate classification ($F_1 > 80\%$) of deviation from a reference posture at thresholds up to 3°, 3° and 2°, respectively. Contact between arm point-features (elbow and wrist) and supporting surface was classified at $F_1 = 80.5\%$. This study demonstrates, for the first time, technical feasibility to automate the identification of (i) a sitting segmental posture including individual trunk segments, (ii) changes away from that posture, and (iii) support from the upper limb, required for the clinical SATCo.

# 1. Introduction

A primary aim of physical therapy for children with neuromotor disability, such as cerebral palsy (CP), is to improve postural control in order to enhance both fine and gross motor skills [1] with postural control of the head and trunk being of primary importance [2]. The accurate assessment of a child's head and trunk control in sitting is thus essential. Current clinical physical therapy assessments of controlled sitting ability for children with CP, although reliable, infer motor control status from the subjective observation of functional abilities [3–6]. The vital clinical need is for an objective measure of seated postural control [7].

A further limitation of these assessments is that they do not consider (i) the multi-segmental nature of the trunk, (ii) any compensatory use of the hands and arms to help maintain an upright posture in the presence of poor head/trunk control, or (iii) give a precise definition of the correct posture. By contrast, the Segmental Assessment of Trunk Control (SATCo) [8] addresses these issues, providing a comprehensive assessment of seated postural control. It systematically assesses control of the neutral vertical posture in sitting at six discrete head/trunk segmental levels and free sitting. The classification of neuromuscular control at a given head/trunk segment requires two conditions to be met: (i) 'alignment': the segment should remain aligned to the neutral vertical posture within a defined threshold, and (ii) 'contact': there should be no contact between the upper limbs and the head/trunk or external surface [8].

Previous work [7,9] has demonstrated that the two components of 'alignment' and 'contact' can be accurately classified from a clinical assessment and quantified using video recordings. However, the methods used previously were semi-automated, requiring manual initialization of postural point-features and necessitating significant human intervention to compensate for tracking drift and occlusion [7,9].

The purpose of this study is to test whether this application is solvable using neural network methods known as 'deep learning'. Our objective is to automate the identification of postural point-features from colour videos of children with cerebral palsy during the crowded environment of a clinical assessment in which the child interacts physically with one or more therapists. The identification of 13 points of interest (2, 6, 2 and 3 points on the head, trunk, pelvis and arm, respectively) is required to estimate the location and orientation of 10 head, trunk and arm segments to automate video-based analysis of seated postural control during the clinical test SATCo.

## 1.1. Overview and justification of the method

Recent developments in machine learning [10–20] justify the value in exploring neural network methods for this application. The goal in image classification is to detect the presence of an object or feature and to achieve invariance with respect to the labels, such that the classification in the final layer of the neural network should remain static in the presence of linear transformation (scale, translation and rotation) of the input image. This can be addressed in neural networks through the use of convolutional layers, which give rise to equivariance in the activations of the neurons, and through the use of max-pooling layers, which achieve local patch (typically $2 \times 2$ patch) invariance to translation [21]. The use of multiple layers of convolution and max-pooling (deep learning) radically improves the neural network's robustness to linear transformation of the input space. However, following this method, much of the precise information about the position of an object of interest within an image is lost [22,23]. To retain the precise detection and tracking of point-features [13,17,24,25], one solution is to use deconvolutional layers and max-unpooling layers [26,27]. These layers, through learning, reverse the process of convolution and recover the position information that was lost during max-pooling, to reconstruct a precise pathway to the full resolution image [12,15,16,28–32]. However, the training time and runtime is relatively expensive. Another possible solution is the use of patch-based convolutional neural network methods [11], but this method has been reported to have problems with fine precision [14]. There are hardware/software and open framework solutions, such as Kinect V1–V2 or OpenPose [33], which provide postural estimation. However, they do not consider the trunk a multi-segment structure, which is essential for the automation of SATCo. Also, existing open frameworks are not validated on children with neuromotor disability, which is essential for this clinical application. More recent sophisticated approaches are based on the concept of adversarial learning [34], but since their relevance is to a more general and complex problem set, they were judged inappropriate as an introductory baseline approach.

Max-pooling is highly cited and recognized as being the main problem to overcome in tasks relating to spatial feature recovery: thus, a simple strategy might apply striding convolutions without pooling, to mimic the same down-sampling effect [35], while retaining high-resolution spatial feature information in

the structure of the network. However, pooling is simultaneously recognized as one of the main strengths of the contemporary deep learning methods. We thus argue that, depending on the domain, a more efficient approach might change the pooling strategy from max-pooling to mean-pooling [36]. Efficient and effective spatial dimensionality reduction is achieved by taking the average of small patches of convolution maps, rather than the max: this retains the possibility, through learning, of conveying precise position information between layers of abstraction via interpolation between adjacent patches, since the pooling function will vary with spatial transformations. Inevitably, the more complicated the task (estimating pose in arbitrary images from the arbitrary viewpoint), the less effective this approach may be. However, since our problem is highly constrained (fixed camera showing the side profile of child), this approach is worth investigating.

Our approach was thus an appropriate deep convolutional architecture based on empirical pre-testing, which used mean-pooling after every convolutional layer, and trained the network with no pre-training or transfer learning, to predict the points of interest of the child's head and trunk directly from individual raw images. For comparison, we trained an equivalent max-pooling network.

# 2. Methods

## 2.1. Participants, procedures and measurements

Data were collected in a previous study [7] from 12 children (9 males and 3 females, mean age $4.52 \pm 2.4$ years, mean height $0.97 \pm 0.1$ m and weight $16.15 \pm 7.5$ kg). All children had a diagnosis of CP and attended a specialist physical therapy centre. The number of sessions recorded per participant varied in relation to their routine attendance during the time the project was running. The inclusion criterion was poor seated head/trunk postural control. The exclusion criteria were fixed bony deformity or other structural problem of the spinal joints and if neither parent/guardian had sufficient understanding of written or spoken English to give informed consent.

Informed written consent was obtained on behalf of their child from a parent or guardian, with written child's assent where possible. Children wore shorts as was usual for their clinical assessments with girls wearing a crop top if required: this allowed accurate palpation of anatomical landmarks for marker placement.

The SATCo was conducted according to the published guidelines [8], assessing control at six trunk segmental levels (head, upper thoracic (UT), mid-thoracic (MT), lower thoracic (LT), upper lumbar (UL), and lower lumbar (LL)), and free sitting, or as far as each child's neuromuscular control permitted, within a single session [7].

Concurrent video was recorded at 25 Hz from a JVC, HD Everio RX110 video camera mounted on a levelled tripod on the right side of the child at a distance of 3.0 m and a height of 0.70 m. This view allowed recording of sagittal plane movements of the head and trunk and arm. Markers (small $2\,cm^3$ coloured bocks) were placed on specific landmarks of the head, trunk and pelvis to improve the lateral visualization and tracking of the back landmarks following the model previously developed (figure 1) [7].

## 2.2. Generation of point-feature labels

Thirty-four sessions were recorded resulting in 177 videos, each of which was processed independently. Three operators, with expertise in labelling, used our custom-created user interface in Matlab to manually annotate 13 point-features from two markers on the head, six markers on the trunk, two markers on the pelvis and one marker on each of the right shoulder, right elbow and right wrist (figure 1). The right arm was the only arm fully visible. This process resulted in 30 825 images, each with an associated set of 13, two-dimensional coordinates describing the points of interest. Where a point-feature was occluded, anatomical landmarks and visible markers, as well as adjacent frames, were used to infer the position of the occluded point-features. The custom user interface superimposed the previous (four frames earlier) annotation, and operators updated the new position of the point-features by clicking nearby. This feature enabled rapid annotation, particularly for sequences or individual point-features which exhibited very little movement. In all sequences, every fourth image was annotated: linear interpolation was used to generate labels for the images without annotations at the completion of manual annotation of the full sequence.

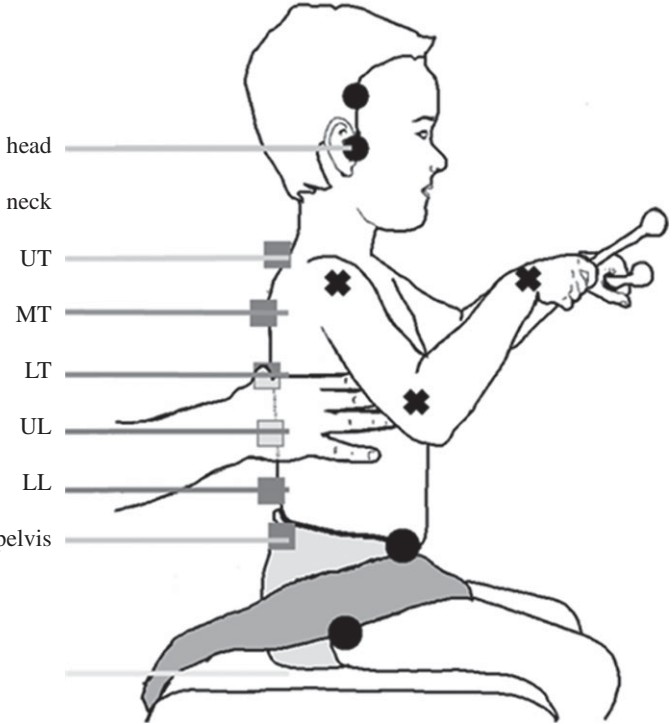

**Figure 1.** Point-feature and segment definitions. Squares show markers placed on the back of the participant: spinous process of the seventh cervical vertebra (C7), third, seventh and 11th thoracic vertebrae (T3, T7 and T11), third lumbar vertebra (L3) and first sacral vertebra (S1). Dots show markers located on the side and front of the child: right ear tragus, right temporal fossa (in a vertical line from the ear tragus when the head was in the neutral position), and greater trochanter and right anterior superior iliac spine (ASIS). Markers define trunk segments: head, neck, upper thoracic (UT), mid-thoracic (MT), lower thoracic (LT), upper lumbar (UL), lower lumbar (LL) and free sitting (pelvis). Crosses show the landmarks tracked for the right arm (shoulder, elbow and wrist), defining segments upper arm (UA) and forearm (FA).

## 2.3. Image processing

Prior to training, all images were cropped to a static region of interest around the children, then resampled using bilinear interpolation to $256 \times 256 \times 3$ pixels. As this dataset was relatively small, some additional processes were added during the training of the networks. To accommodate natural sagittal shift of the children relative to the bench and camera, the input images were randomly translated, sampling horizontal and vertical shift parameters from a real uniform distribution (single precision) in the range of [−16 16] (pixels): this was repeated for each learning iteration (a forward pass, an error calculation, a backward pass and a weight/parameter update). The labels (point-feature coordinates) were augmented with the sampled translation parameters before each learning step. Bilinear interpolation was used to sample the pixel intensities during transformation. The conversion of pixels to metric space was not possible with these data, and thus, labels were delivered in the pixel space. Following translation, local contrast normalization was applied via the graphics processing unit (GPU; runtime less than 0.001 s) with a local neighbourhood of $19 \times 19$ (pixels): this accounted for natural variation in contrast, brightness and skin tone. These two processes, respectively, artificially increased the size of the training dataset and increased the speed of convergence: both improving processes also improved generalization to the testing dataset by limiting over-fitting.

## 2.4. Network architecture

The network architecture selected was a moderately deep, five convolutional and mean-pooling layers, consisting of [64, 64, 128, 128, 256] (from high spatial resolution to low spatial resolution) exponential linear units (ELUs), three nonlinear fully connected layers of 512 ELUs and a final linear regression layer of 26 units ($13x + 13y$). The first convolutional layer used strides of $2 \times 2$, while all other convolutional layers used strides of $1 \times 1$. The spatial filter sizes in each convolutional layer were [$9 \times 9 \times 3$, $5 \times 5$, $5 \times 5$, $3 \times 3$, $3 \times 3$] (from high spatial resolution to low spatial resolution). All pooling layers operated on neighbourhoods of $2 \times 2$, down-sampling convolution maps by a factor of 2.

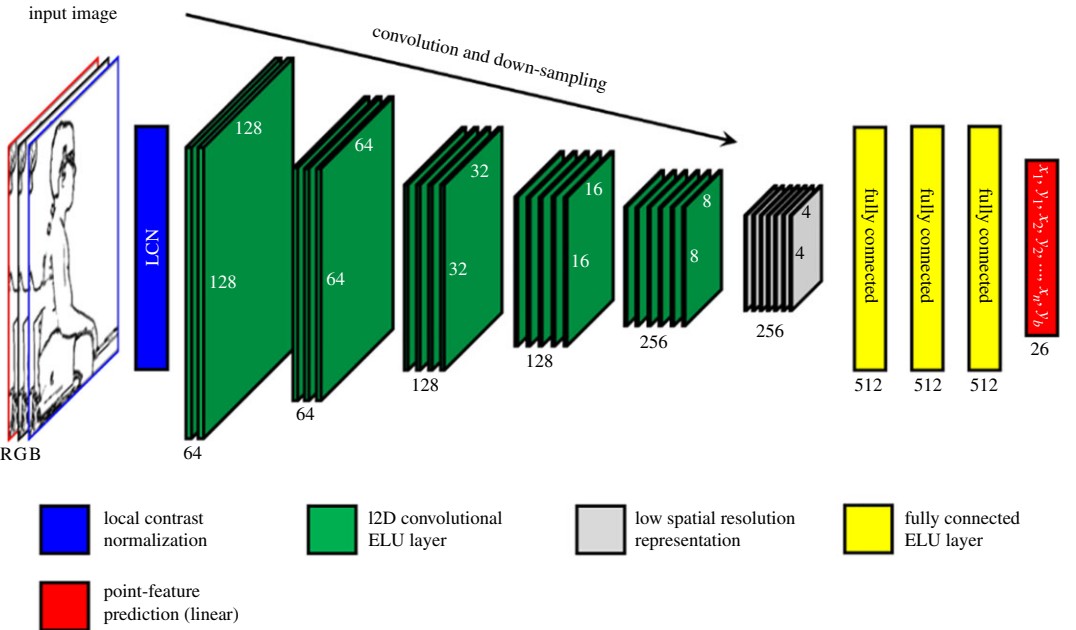

**Figure 2.** Neural network architecture. Layers of convolutional ELU filters (green) are shown with spatial down-sampling (average-pooling) to learn a low-resolution spatial feature representation of the $256 \times 256 \times 3$ (RGB pixels), local contrast normalized (LCN) (blue), input image* (far left). Multiple fully connected ELU layers (yellow) then map that feature representation to the prediction of 13 points of interest, with a final linear regression layer (red). *Actual full-colour image cannot be shown for ethical reasons; a cartoon/edge image is shown to illustrate the process.

After five layers of convolution and pooling, the image representation layer had a spatial resolution of only $4 \times 4$ pixels with 256 nonlinear features. Down-sampling was not taken any further in order that some spatial detail was retained (figure 2). Each network was trained on approximately 28 000 training images for approximately 500 000 learning iterations (18 cycles through the training set). Two different networks were trained in parallel on different GPUs.

All networks were initialized according to the following scheme based on the literature and our experience of training neural networks. Linear unit weights were drawn from a real (single precision) uniform distribution in the range $\left[ -\sqrt{3/\text{fan in}} \quad +\sqrt{3/\text{fan in}} \right]$. ELU unit weights were drawn from a real (single precision) uniform distribution in the range $\left[ -\sqrt{6/\text{fan in}} \quad +\sqrt{6/\text{fan in}} \right]$, where the *fan in* is the total number of the local (spatial) and feature inputs to any given unit in a convolutional layer and the total number of feature inputs to any given unit in a fully connected layer.

## 2.5. Deep learning software

All neural network software was developed entirely within the group at Manchester Metropolitan University, with all code/software written solely by the first author using C/C++ and CUDA-C (Nvidia Corporation, Santa Clara, CA, USA). Only the standard CUDA libraries (runtime v. 8.0 cuda.h, cuda_runtime.h, curand.h, curand_kernel.h, cuda_occupancy.h and device_functions.h) and the C++ 11 standard library were used. All 12 neural networks were trained on an AMD Athlon X4 860 K Quad-Core 3.7 GHz CPU, 32 GB (2400 MHz), with two Nvidia GTX 1080 GPUs.

## 2.6. Training and cross-validation

Adaptive moment estimation (ADAM) was used with default $\beta_1 = 0.9$ and $\beta_2 = 0.999$ parameters, but with a smaller ($\alpha = 0.0001$) than suggested $\alpha = 0.002$ parameter (learning rate) to account for non-batch (batch size of 1) learning, to provide optimal generalization/testing performance [37–39]. All parameters were empirically selected using a subset of the data to check for quick (with respect to weight updates) and stable (no 'exploding gradients') convergence.

Cross-validation was executed with 12 folds, where, for each set of network parameters and properties (architecture/units), 12 identical networks were separately trained using 11 of the 12 participants' images

and labels to train each network and the remaining participant's images and labels to test each network (split over the trials 50% validation and 50% testing). Participant's images and labels used to test any of the 12 total networks were not used to test any of the other 11 networks. This process yielded genuine held-out test results for all 12 participants.

The mean absolute error (MAE) was minimized through online learning, which was interrupted every quarter pass through the training set (approximately 7000 learning iterations), to record MAE test results from the two test sets, individually. If the MAE for either test set was lower than any previously recorded MAE for that test set, the network was saved to long-term storage. When neither test set recorded a lower MSE for 32 consecutive test iterations, training was terminated. At the end of training, the network associated with the lowest loss for test set 1 was loaded to acquire results for test set 2, and vice versa, yielding true held-out optimal results for both test sets.

## 2.7. Classification of head/trunk segment 'alignment'

After network training and production of held-out test output for all 12 children, all point-feature data were used to calculate the segment angles of the head, neck and trunk segments as required for SATCo [7]. The SATCo classifies a segment with absolute deviation more than 20° from the reference aligned posture as 'misaligned' [8]. Here, for this test of feasibility, the reference posture was defined per child as the mean posture over all levels and sessions, as defined by the labels, not by the neural network. For each of eight head–trunk segments, the label-derived angles and neural network-derived angles were thresholded at a range of angles (±1 to ±40°) and classified as positive when above the threshold. Agreement between the label and network-derived classification was measured using accuracy, true positive rate, false positive rate, precision, recall and the $F_1$-score (table 2). Since the class populations are severely unbalanced, precision (true positive/total predicted positive), recall (true positive/total actual positive) and specially $F_1$ (harmonic mean of precision and recall) provide the most valuable measures.

## 2.8. Classification of upper limb 'contact'

A fixed threshold was defined which, for any given image, was the position of the upper pelvis marker (anterior superior iliac spine (ASIS)) as defined by the labels, and not by the neural network. Then, for both neural network-derived point-features and for label-derived point-features, the position of the elbow marker was positively classified as in contact with external support if it was to the left of the ASIS marker, and the position of the wrist was classified as in contact with external support if it was below the ASIS marker.

# 3. Results

## 3.1. Image-relative point-feature localization

The labels and the neural network output comprise 26 real-valued numbers defining $x$ and $y$ coordinates of the 13 point-features. The neural networks were trained to minimize the absolute difference between their output and the labels. Results, in terms of difference in pixels, showed that overall the neural network was able to locate each point-feature to within 4.40 pixels ±3.75 (figure 3). Image resolution was 256 × 256 pixels.

## 3.2. Segment angles

To generate results comparable to the existing literature, segment angles were computed according to [7,9]. Results for the relevant segments (head, neck, UT, MT, LT, UL, LL and pelvis) for our fully automated method recorded comparable performance, 6.4° (MAE), to the current benchmark semi-automated method, 3.8° (MAE) (table 1). Both neural networks compared very well with the benchmark, particularly in the trunk segments (LL-UT), where the optimal MT error was actually given by the neural network methods and not Dartfish. The segments most contributing to the error were the head, upper arm, forearm, lower lumbar and pelvis, with errors between 8.5° and 9.9° (MAE), whereas the trunk and neck segments gave errors between 4.2° and 6.2° (RMSE). Within the trunk, the mid-trunk segments recorded strongest performance (UL, LT and MT). Comparison

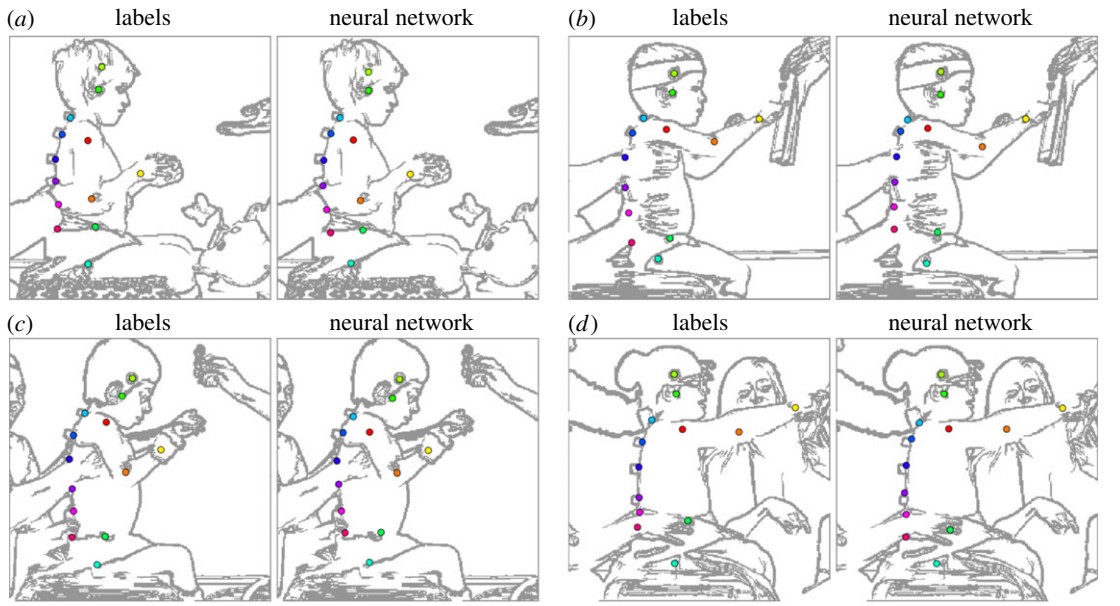

**Figure 3.** Neural network visual comparison with labels, showing representative held-out test results of neural network performance compared with the manual labels. The images* show sagittal views from one image in a video sequence of child sat on a bench in the centre of the image, and one or more physiotherapists performing the SATCo test at a variety of segmental levels. Coloured circles mark the locations of the 13 point-features. There are four pairs of images, showing the labels (left image) and the neural network prediction (right image) from the same raw colour image. All four examples illustrate high prediction accuracy in the presence of occlusion, particularly trunk (*a*), lower thoracic and upper/lower lumbar trunk (*b*), mid-thoracic, upper/lower lumbar trunk (*c*), mid-thoracic, lower thoracic trunk (*d*), upper/lower lumbar trunk. *The actual colour images used in the analysis cannot be shown for ethical reasons; therefore, cartoon images are shown to illustrate.

between neural network methods, mean-pooling and max-pooling revealed marginally stronger performance for the mean-pooling method, with errors over all segments of 6.8° compared with 6.9°. Notable differences were in arm and head segments (figure 4).

## 3.3. Classification of 'alignment'

For a fixed threshold of 20°, accuracy for segments ranged from 84.0 to 99.6% (table 2). The number of positive classifications in the labels (misaligned) ranged from 147 (UL) to 5003 (head). Accounting for the high imbalance in positive and negative classes, precision and recall measures were computed for UL, MT, UT, neck and head segments. The other segments did not contain enough cases to representatively summarize performance. Precision ranged from 0.8 to 100.0%. Recall ranged from 2.7 to 61.4%. False positive rates were low, ranging from 0.0 to 9.2%. False negatives were the main source of error, with values ranging from 38.6% (neck) to 99.9% (LL). $F_1$ ranged from 0.2% (UT) to 63.7% (head).

For all segments, $F_1$ was very high for low thresholds and decreased as threshold increased (figure 5). For the six trunk segments (UT, MT, LT, UL, LL and pelvis), neural network prediction was reliable ($F_1 > 0.5$) for thresholds up to 4–5°. For the neck and head, prediction was reliable ($F_1 > 0.5$) for thresholds up to 14° and 12°, respectively.

## 3.4. Classification of upper limb 'contact'

From the labels, the ASIS marker defined dynamic lateral and vertical thresholds linked to the pelvis, with which to detect contact of the arm with the trunk or support surface. This analysis revealed very high precision and recall rates of 85.7% and 76.0%, respectively, with an accuracy of 93.3% (table 3).

Analysis of the individual point-features of elbow and wrist revealed the elbow position to be the most reliable point-feature in determining contact with the body (table 3). The elbow recorded accuracy at 96.3% with 76.0% true positive classification and only 11.2% false negative classification, while the wrist recorded 20.5% and 79.5% true positive and false negative, respectively. The $F_1$-score was significantly higher in the elbow than the wrist at 87.3% compared with 30.6%. Electronic supplementary material, figure S4 shows a representative example of this process.

**Table 1.** Neural network comparison to benchmark. Results are presented in the form of MAE of the difference between predicted (neural network) or tracked (benchmark) segment angles and manually annotated segment angles. The italic in the Dartfish column denotes the best results over all three methods. The italic in the neural network columns denote the best result over both neural network methods. The table shows that the neural network gave comparable performance with the benchmark, mean-pooling was slightly better than max-pooling, and for the MT segment both neural networks outperformed the benchmark significantly. Benchmark results do not include any occluded features, whereas the neural network results do. The benchmark was not capable of reliably tracking the arm segment.

| | Dartfish | neural network | |
| --- | --- | --- | --- |
| | Sánchez et al. [7] | mean-pooling | max-pooling |
| upper arm | — | *6.7 ± 9.5* | 7.5 ± 10.6 |
| forearm | — | *9.9 ± 12* | 10.7 ± 12.1 |
| head | *4.5 ± 7* | 9.6 ± 8.9 | 8.7 ± 8.3 |
| neck | *2.3 ± 3.3* | 4.8 ± 6.1 | 5 ± 5.9 |
| UT | *5.3 ± 9.8* | 6.2 ± 5.8 | 6.2 ± 6 |
| MT | 4.3 ± 11.3 | 4.2 ± 3.3 | *4.1 ± 3.2* |
| LT | *3.2 ± 4.5* | 5.1 ± 4.1 | *5 ± 4.1* |
| UL | *3.8 ± 6.3* | 4.4 ± 3.4 | 4.8 ± 3.6 |
| LL | *7 ± 10.2* | 8.7 ± 7.4 | 8.9 ± 7 |
| pelvis | *3.7 ± 7.9* | 8.5 ± 7.4 | 8.6 ± 7.3 |
| arm[a] | — | *8.3 ± 8.9* | 9.1 ± 9.4 |
| trunk[b] | *3.7 ± 5.1* | 5.6 ± 2.6 | 5.7 ± 2.4 |
| reference[c] | *3.8 ± 5.2* | 6.4 ± 2.8 | 6.4 ± 2.7 |
| all segments[d] | *3.8 ± 5.2* | 6.8 ± 3.1 | 6.9 ± 3.1 |

[a]Combined results of upper arm and forearm.
[b]Combined results of UT, MT, LT, UL and LL.
[c]Combined results of head, neck, UT, MT, LT, UL and LL.
[d]Combined results of all segments.

## 4. Discussion

The purpose of this study was to test whether the estimation of 13 postural point-features from colour videos of children with cerebral palsy was feasible using neural network methods. Estimation should be robust during the crowded clinical environment of a SATCo test. The method should detect 'alignment', i.e. deviation of each of eight head/trunk segments from a reference orientation, and 'contact' of the upper limb with the trunk or support surface to the accuracy required to fully automate video-based SATCo analysis of seated postural control in children with neuromuscular disorders.

Deep convolutional neural networks were applied to an existing video dataset of SATCo assessments of children with cerebral palsy. This same dataset had been used in previous research to develop a semi-automated, objective method for quantifying postural control [7]. That previous method reported problems of tracking drift and feature dropout (loss of feature during tracking), due to three-dimensional motion in the two-dimensional image plane, as well as occlusion of point-features [7,9]. The present study introduced deep learning methods to overcome these problems and provide a fully automatic approach, which predicts the point-features directly from single images (i.e. no tracking and no initialization).

We compared our deep learning methods with the benchmark feature tracking method, Dartfish [7], by measuring MAE between method-derived segment angles and label-derived segment angles. The two neural network methods, max-pooling and mean-pooling, performed similarly, but in almost all segments the mean-pooling method gave superior performance. With these qualifications in mind, the benchmark method demonstrated higher tracking accuracy than the neural network methods, in all segments except the mid-thoracic trunk segment (table 1)—an area in the trunk commonly occluded by physiotherapists in the scene. It is noteworthy that the benchmark method was expected to give

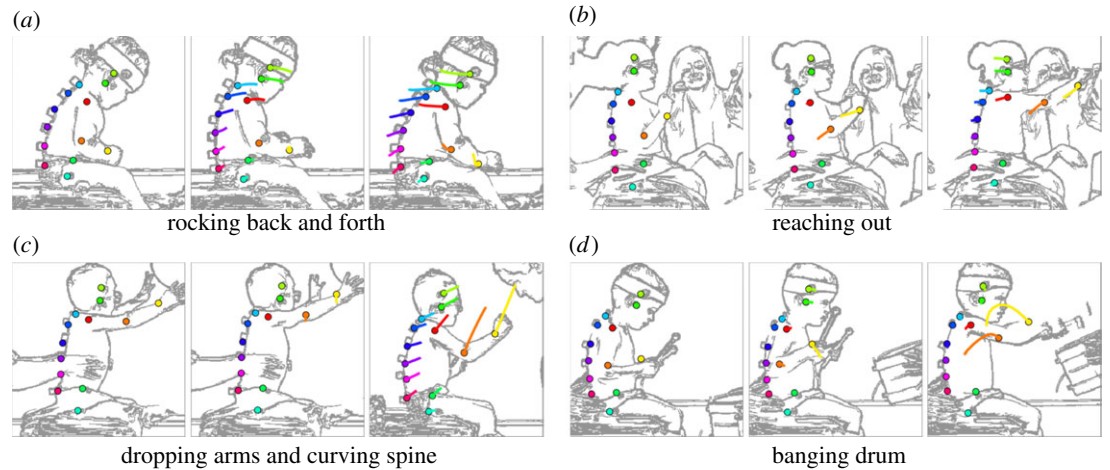

**Figure 4.** Illustrative neural network motion tracking results. This figure shows the stability of the neural network predictions over a set of selected actions for four representative participants. Each of the four panels shows three images*, the leftmost image is the start of the action sequence, while the rightmost image is the end of the action sequence (30 frames after the start), and the centre image is the middle of the sequence (15 frames after the start and 15 before the end). The coloured dots represent the point-features predicted by the neural network at the current frame, while the trailing coloured lines represent the historical path of the point-features predicted by the neural network in the previous 10 frames (0.4 s). While there are some discrepancies, it appears that each action sequence has been captured by the neural network. *The actual colour images used in the analysis cannot be shown for ethical reasons; therefore, cartoon images are shown to illustrate.

the strongest performance because every time the feature points drifted or dropped out, the operator recovered the point manually; therefore, it was certain to be close to manual labels. The upper limb segments were not addressed by the benchmark method. However, the benchmark method requires initialization and manually supervised tracking of every point-feature in the head, trunk and pelvis, whereas the proposed method locates all point-features, including additional three point-features on the right arm directly from the raw image, with no user interaction.

Machine learning methods require data to learn and this dataset is very small, with only 12 children, of whom only three are female. It is expected that additional data would improve substantially the performance of the neural network trained in this study. Additionally, encouraged by the results of this study, exploration of many available suitable methods [10–20], including repurposing existing models like OpenPose [33], offers the prospect of improving performance independently of dataset size.

To translate the accuracy of this neural network method into measures relevant to SATCo testing, we assessed ability to classify 'alignment' and 'contact' as the two processes required by the SATCo test to determine, for each head/trunk segment tested, whether neuromuscular control is demonstrated.

'Alignment': The angular deviation of a segment from a reference orientation, greater than a threshold (e.g. $\pm 20°$ [8]), is classed as positive, i.e. 'misaligned'. During clinical SATCo, the reference orientation is alignment to the neutral vertical posture. This study is testing feasibility to estimate point-features and not testing feasibility of training a neural network to identify the neutral aligned vertical posture. Thus, we used the mean posture per child from the annotated labels as the reference orientation for both the label and the network-predicted alignment. The calculation of segment angles from point-features was fully automatic.

Agreement between labels and predicted alignment for head and neck segments was mainly correct ($F_1 > 50\%$) at thresholds up to the values used by clinicians $\pm 20°$ [8] and $\pm 17°$ [7] (electronic supplementary material, figure S3). For the lower segments (UT, MT, LT, JL, LL and pelvis), agreement between labels and predicted alignment was usually correct ($F_1 > 50\%$) at a lower range of thresholds up to 5° (electronic supplementary material, figure S3).

In clinical testing, a child may have ability to control a segment and yet not demonstrate that ability, e.g. because the child is tired, cannot be bothered or does not want to play. The lack of demonstration of control proves little. However, if control is demonstrated, one demonstration is sufficient to establish that control is possible. For clinical testing, the error that should be minimized is to predict alignment when the child is misaligned (false negative), and the complement to be maximized is true classification of misalignment (true positive).

The main contributor to disagreement between labels and the neural network on the detection of misalignment was the false negative rate, i.e. segments classified as aligned which are actually misaligned. The mean neural network false negative rate was approximately 76.7%, while the false

**Table 2.** Neural network-predicted segment angle threshold analysis results. Results are presented in various forms of accuracy measure, on classification of segment alignment based on neural network and label-derived segment angles. A fixed threshold of ±20° was used to classify the labels. Low numbers of positive classes lead to poor estimates of performance. In the MT, UT, neck and head segments, ample positive classifications provide good estimates of performance. False negative rates are the main source of error.

| segment | label threshold | neural network threshold | no. of positive classes | no. of negative classes | accuracy | true positive rate (recall) | false positive rate | true negative rate | false negative rate | precision | $F_1$-score |
|---|---|---|---|---|---|---|---|---|---|---|---|
| head | 20° | 20° | 5003 | 25 822 | 84.0% | 48.7% | 9.2% | 90.8% | 51.3% | 50.7% | 49.7% |
| neck | 20° | 20° | 2074 | 28 751 | 95.3% | 61.4% | 2.3% | 97.7% | 38.6% | 66.3% | 63.7% |
| UT | 20° | 20° | 959 | 29 866 | 96.2% | 23.3% | 1.5% | 98.5% | 76.7% | 33.7% | 27.5% |
| MT | 20° | 20° | 296 | 30 529 | 99.2% | 39.2% | 0.2% | 99.8% | 60.8% | 65.5% | 49.0% |
| LT | 20° | 20° | 208 | 30 617 | 99.3% | 7.2% | 0.1% | 99.9% | 92.8% | 39.5% | 12.2% |
| UL | 20° | 20° | 147 | 30 678 | 99.5% | 2.7% | 0.0% | 100.0% | 97.3% | 100.0% | 5.3% |
| LL | 20° | 20° | 909 | 29 916 | 96.6% | 0.1% | 0.4% | 99.6% | 99.9% | 0.8% | 0.2% |
| pelvis | 20° | 20° | 1102 | 29 723 | 95.6% | 3.4% | 1.0% | 99.0% | 96.6% | 11.8% | 5.3% |

**Table 3.** Neural network classification of external support via the arm. Results are presented on the accuracy of the neural network in predicting contact of the arm with a supporting structure (the child's own body, somebody else, the bench or some other object). Classification rates in the form of accuracy, recall, precision and (or $F_1$-score) are very high. The elbow point-feature gives the highest true positive and the lowest false negative rate of classification, while the wrist point-feature gives the highest precision. False negative rate in both sets of analysis are the main source of error.

| point-feature (s) | no. of positive classes | no. of negative classes | accuracy | true positive rate | false positive rate | true negative rate | false negative rate | precision | $F_1$-score |
|---|---|---|---|---|---|---|---|---|---|
| elbow and wrist | 5583 | 25 242 | 92.1% | 69.2% | 2.9% | 97.1% | 30.8% | 84.2% | 76.0% |
| elbow | 4458 | 26 367 | 94.5% | 77.9% | 2.7% | 97.3% | 22.1% | 83.2% | 80.5% |
| wrist | 1445 | 29 380 | 96.0% | 24.2% | 0.5% | 99.5% | 75.8% | 71.4% | 36.1% |

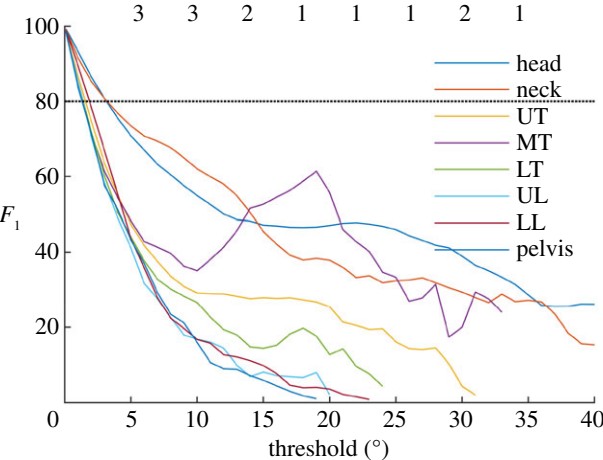

**Figure 5.** Variation of $F_1$ with threshold. For all segments (except the arms), $F_1$ shows the agreement between neural network-derived and label-derived classification of 'alignment' for a range of thresholds ±1° to ±40°.

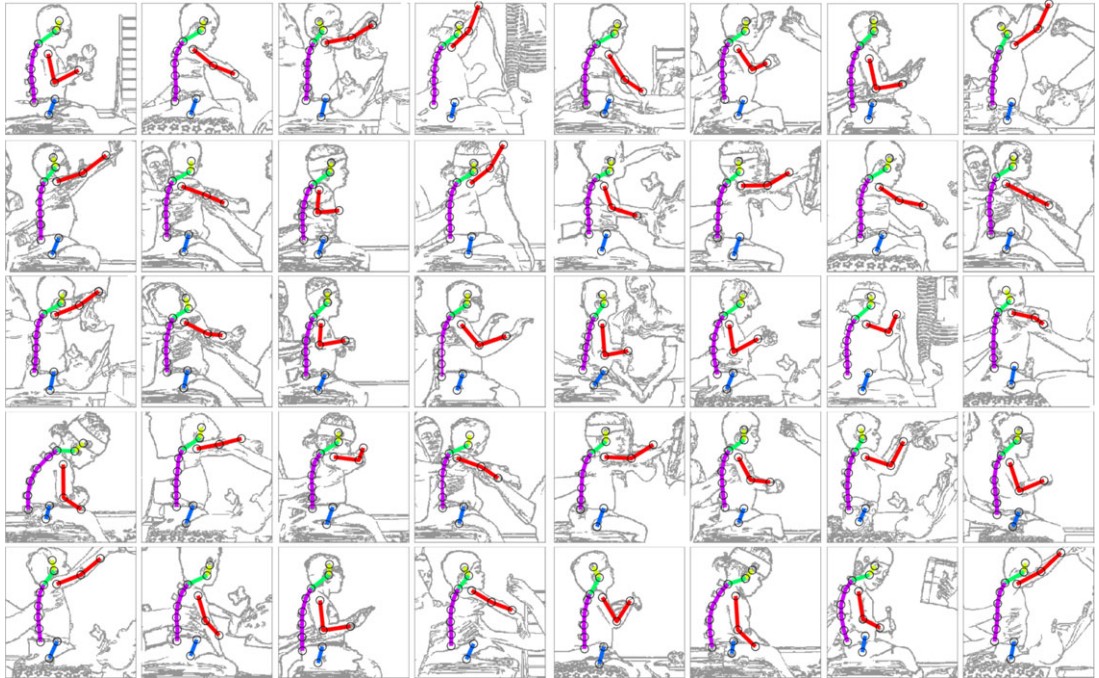

**Figure 6.** Forty randomly selected example neural network point-feature predictions. This figure shows images* and neural network predictions for a range of possible poses, randomly selected from all 30 825 images. Each of the 40 panels show an image of a child, and the neural network prediction grouped by the segment (head = yellow, neck = green, arm = red, trunk = purple, hip = blue). This figure is representative of the neural network's ability to capture the pose of a child it has not been trained on. Ground truth is not shown to aid clear inspection, but predictions which do not match the image entirely are evident (e.g. the arm in row 5, column 2, and the head in row 5, column 7). In general, the neural network appears robust to new data and new poses. *The actual colour images used in the analysis cannot be shown for ethical reasons; therefore, cartoon images are shown to illustrate.

positive rate was very low (average: 1.8%). The higher false negative rates lie in the lower segments and are related to the substantially fewer cases of misalignment (no. of positive classes, table 2). Datasets containing more children lacking control of the lower trunk segments need to be collected.

Results from this system can be visually verified by seeing if the predicted pose matches the image and the video sequence. Generally, visual inspection (figure 6; electronic supplementary material, media) shows that the results are good.

'Contact': The threshold marker (ASIS) was defined only by the labels to provide for us, rather than the clinical user, the most accurate assessment of contact relative to a known external location. The results (table 3) showed that 'contact' could be detected with high accuracy, relative to the amount of data

available, with a very high $F_1$-score of 80.5% (electronic supplementary material, figure S4). The arm had no physical markers (small blocks) and, if present, these would have helped the neural network to locate the point-features. Nevertheless, the neural network was able to locate the elbow and the wrist in a given image in the absence of artificial visual clues, with sufficient accuracy to detect contact with an arbitrary body.

Visual inspection (see electronic supplementary material, videos) of some of the neural network predictions indicated that where the left arm was, in view, above or below the right arm, the network became confused about the position of the wrist, predicting a point approximately in the average location between the left and right wrist. In principle, this error would be addressed with more data giving more training examples to discriminate. In future work, we propose using an 'RGB-D' camera. The addition of depth data to complement the colour images would remove the ambiguity about the left and right arm in the two-dimensional sagittal plane.

The results of this study justify further development including the collection of additional data and the application and tuning of more sophisticated and established techniques, such as the Mask R-CNN [19] or the adversarial PoseNet [34]. Our experience suggests that there is no necessity to replicate the same volume of labels per additional child in the dataset. Performance would probably increase if additional variation in the labels and images was introduced in the form of additional children. The current study generated over 30 000 labels. The generation of 30 000 labels from more children would add useful variation in body shape and type, developmental age and the severity of disability while retaining the intraparticipant variation in the range of motion per segment. From these results, we estimate that between 30 and 50 children, each with 1000 annotations would result in a strong dataset for the delivery of a highly accurate neural network analysis. If the number of children was increased to around 100, with the number of labelled images around 100 000, this would allow for larger cross-validation testing batches, reducing the bias when regularizing by early stopping of training of the neural network.

# 5. Conclusion

This study demonstrates, for the first time, the technical ability to automate the identification of (i) a posture comprising individual segments, (ii) changes away from that posture, and (iii) support from the upper limb to a level of accuracy and sensitivity comparable to the current SATCo clinical standard [8] in children with cerebral palsy. The application of our method can be widened to include other participant groups, such as typically developing infants and children with neuromuscular disease. Our method applied a standard convolutional architecture, using mean-pooling to retain some ability for point-feature localization. It has potential to be accessible and reproducible by other laboratories worldwide through the use of publicly available code repositories. We have demonstrated high levels of performance ($F_1 > 50\%$), through state-of-the-art cross-validation, on a small dataset of only 12 children: this proves efficacy and feasibility for expansion into larger sample sizes. We have also demonstrated that labelling data is straightforward. Our results justify the collection of additional data that is designed for purpose (i.e. no markers on the child), the investigation and comparison of a wider range of more sophisticated deep learning methods and the introduction of depth information to complement the colour images. This will enable a fully automated assessment to address the vital clinical need for an objective measure of seated trunk control and to enhance planning and monitoring of a wide variety of interventions in children with neuromotor disability.

Ethics. Ethical approval for the study (REC reference no. 14/SC/1182) was obtained from the NHS Health Research Authority (NRES Committee South Central, UK) and from the University Ethics Committee. The study was conducted in accordance with the Declaration of Helsinki guidelines.

Data accessibility. This manuscript is accompanied by electronic supplementary material media (videos) and a document. The document contains five figures, giving more detail on the videos, neural network feature maps, training and testing error curves for all neural networks, representative neural network time series showing the classification of segment alignment and representative neural network time series showing the classification of external support via the arm.

Authors' contributions. This study was planned, and the manuscript was prepared by all authors. Data collection was conceived and conducted by P.B.B., M.B.S. and I.D.L. Annotation and deep learning code/analysis was written by R.C. Analysis was performed by R.C. and I.D.L. Labelling was performed by R.C., M.B.S. and M.J.S. All authors interpreted analysis and results. All authors gave final approval for publication.

Competing interests. We declare we have no competing interests.

Funding. We received no funding for this study.

Acknowledgements. The authors express sincere thanks to the children whose data were used in this study, their families and to the staff of The Movement Centre, Oswestry, UK.

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
