## [Reviewer comments · Royal Society Open Science]

Review History

RSOS-191011.R0 (Original submission)

Review form: Reviewer 1

Is the manuscript scientifically sound in its present form?

Yes

Are the interpretations and conclusions justified by the results?

Yes

Is the language acceptable?

Yes

Do you have any ethical concerns with this paper?

No

Have you any concerns about statistical analyses in this paper?

No

Recommendation?

Accept as is

Comments to the Author(s)

The paper presents good results and an effective objective measurement tool.

Review form: Reviewer 2

Is the manuscript scientifically sound in its present form?

Yes

Are the interpretations and conclusions justified by the results?

Yes

Is the language acceptable?

Yes

Do you have any ethical concerns with this paper?

No

Have you any concerns about statistical analyses in this paper?

No

Recommendation?

Accept with minor revision (please list in comments)

Comments to the Author(s)

This paper proposed a fully automated deep learning based posture estimation method for predicting body key points of children with cerebral palsy. The input to this method is an image and the output is 13 2D points.

The research problem is significant and interesting. The experiments are thorough and evaluates several aspects of the proposed method and compares with methods from the literature. The results are not significant but justified by the authors.

I have few comments to the authors:

1. Why the authors did not use one of the standard, pre-trained deep neural networks such as ResNet-18?
2. There is no significant difference between using max or mean pooling, and also I believe the computational cost is almost identical. So, in my view this sentence is confusing. [49] Mean-pooling provides a comparatively simple architecture and therefore higher runtime efficiency.
3. why did the authors use a batch size of 1 sample, please justify.
4. The weight initialisation method is not clear, please elaborate more.
5. Section F, too technical details that are not helpful, and why not use something like Tensorflow or PyTorch which are standard deep learning software libraries?
6. Have the authors considered retraining OpenPose for children pose estimation?

Decision letter (RSOS-191011.R0)

05-Aug-2019

Dear Dr Cunningham,

The editors assigned to your paper ("Fully automated image-based estimation of postural point-features in children with cerebral palsy using deep learning") have now received comments from reviewers. We would like you to revise your paper in accordance with the referee and Associate Editor suggestions which can be found below (not including confidential reports to the Editor). Please note this decision does not guarantee eventual acceptance.

Please submit a copy of your revised paper before 28-Aug-2019. Please note that the revision deadline will expire at 00.00am on this date. If we do not hear from you within this time then it will be assumed that the paper has been withdrawn. In exceptional circumstances, extensions may be possible if agreed with the Editorial Office in advance. We do not allow multiple rounds of revision so we urge you to make every effort to fully address all of the comments at this stage. If deemed necessary by the Editors, your manuscript will be sent back to one or more of the original reviewers for assessment. If the original reviewers are not available, we may invite new reviewers.

- Data accessibility

If you wish to submit your supporting data or code to Dryad (<http://datadryad.org/>), or modify your current submission to dryad, please use the following link:
<http://datadryad.org/submit?journalID=RSOS&manu=RSOS-191011>

- **Competing interests**

- **Authors' contributions**

- **Acknowledgements**

- **Funding statement**

Best regards,

on behalf of Marta Kwiatkowska (Subject Editor)
openscience@royalsociety.org

Associate Editor's comments to the Author(s):

The reviewers have provided some feedback that the editors would like you to address. Please ensure you fully respond to their commentary, and provide a point-by-point response to their critiques when you resubmit.

Reviewers' Comments to Author:

Reviewer: 1

Comments to the Author(s)

The paper presents good results and an effective objective measurement tool.

Reviewer: 2

Comments to the Author(s)

This paper proposed a fully automated deep learning based posture estimation method for predicting body key points of children with cerebral palsy. The input to this method is an image and the output is 13 2D points.

The research problem is significant and interesting. The experiments are thorough and evaluates several aspects of the proposed method and compares with methods from the literature. The results are not significant but justified by the authors.

I have few comments to the authors:

1. Why the authors did not use one of the standard, pre-trained deep neural networks such as ResNet-18?
2. There is no significant difference between using max or mean pooling, and also I believe the computational cost is almost identical. So, in my view this sentence is confusing. [49] Mean-pooling provides a comparatively simple architecture and therefore higher runtime efficiency.
3. why did the authors use a batch size of 1 sample, please justify.
4. The weight initialisation method is not clear, please elaborate more.
5. Section F, too technical details that are not helpful, and why not use something like Tensorflow or PyTorch which are standard deep learning software libraries?
6. Have the authors considered retraining OpenPose for children pose estimation?

Author's Response to Decision Letter for (RSOS-191011.R0)

See Appendix A.

RSOS-191011.R1 (Revision)

Review form: Reviewer 2

Is the manuscript scientifically sound in its present form?

Yes

Are the interpretations and conclusions justified by the results?

Yes

Is the language acceptable?

Yes

Do you have any ethical concerns with this paper?

Yes

Have you any concerns about statistical analyses in this paper?

No

Recommendation?

Accept as is

Comments to the Author(s)

Thank you for addressing the comments and your justifications.

Decision letter (RSOS-191011.R1)

23-Sep-2019

Dear Dr Cunningham,

I am pleased to inform you that your manuscript entitled "Fully automated image-based estimation of postural point-features in children with cerebral palsy using deep learning" is now accepted for publication in Royal Society Open Science.

Please note that 'M.Southgate@mmu.ac.uk' is not currently receiving messages from the journal (an error message indicates a possible spam filter is active). Please can you confirm with the above journal email addresses an active email for Dr Southgate?

on behalf of Prof Marta Kwiatkowska (Subject Editor)
openscience@royalsociety.org

Reviewer comments to Author:
Reviewer: 2

Comments to the Author(s)
Thank you for addressing the comments and your justifications.

Appendix A

Associate Editor's comments to the Author(s):

The reviewers have provided some feedback that the editors would like you to address. Please ensure you fully respond to their commentary, and provide a point-by-point response to their critiques when you resubmit.

Reviewers' Comments to Author:

Reviewer: 1

Comments to the Author(s)

The paper presents good results and an effective objective measurement tool.

Thank you for this comment, we find the results very encouraging and supportive of further development within this domain.

Reviewer: 2

Comments to the Author(s)

This paper proposed a fully automated deep learning based posture estimation method for predicting body key points of children with cerebral palsy. The input to this method is an image and the output is 13 2D points.

The research problem is significant and interesting. The experiments are thorough and evaluates several aspects of the proposed method and compares with methods from the literature. The results are not significant but justified by the authors.

I have few comments to the authors:

1. Why the authors did not use one of the standard, pre-trained deep neural networks such as ResNet-18?

Thank you for this question. This paper does not test whether pretrained models can be used to solve pose estimation in children with disability via RGB images. Rather, "the purpose of this study is to test whether this application is solvable using neural network methods" (quoted from final paragraph of introduction).

Our results make a contribution to the field, in that a plain convolutional neural network, trained from scratch, can predict pose from regular RGB images of children with cerebral palsy, during SATCo testing – that was previously not known. It would perhaps be assumed by the community that pretrained models, or more complicated models would be required – we have demonstrated otherwise.

We agree that pretrained models usually provide improved performance where the dataset is small. However, we believe the more valuable investigation is in the architectures (see 'overview and justification of methods') – something which is discussed in this paper, but not tested. Our results are satisfactory and support the acquisition of significant quantities of data, and investigation of different architectures on those data, possibly including repurposing existing models in the public domain.

2. There is no significant difference between using max or mean pooling,

Thank you, yes, we agree, our findings as reported in the manuscript, show that mean pooling offered a notable but not significant improvement over max pooling.

and also I believe the computational cost is almost identical. So, in my view this sentence is confusing. [49] Mean-pooling provides a comparatively simple architecture and therefore higher runtime efficiency.

Thank you for pointing this out. You are correct, before our original submission, our implementation of max pooling was not atomised, but our implementation of mean pooling was. Our current implementation shows that they have equivalent runtime cost, but actually training mean pooling models is negligibly slower due to the propagation of gradients to 4 points in the 2x2 mean pool, as opposed to only 1 point in the 2x2 max pool. This may not be the case with other frameworks. Either way, you are correct, and we have removed the comment from the manuscript.

3. why did the authors use a batch size of 1 sample, please justify.

Thank you for pointing this out, we did not include a supporting reference, which is now added to the manuscript [1], [2] (see table 2, 'testing accuracy' in [2]).

This topic is currently being debated amongst the neural networks/deep learning community and is certainly not settled. Large batch (>1) training is far more efficient than online (batch = 1) training on current hardware [2], [3], though efforts are being directed to optimize small batch training due to the gain in generalisation performance.

The consensus currently lies with small batches (<=5) generalise to test cases better than large batches, but there is sparse evidence which may suggest batch size of 1 is 'better' than batch size of 5.

In our future work, we will consider different batch sizes, however, there is negligible justification for executing slightly larger batch (<=5) learning within this study, since there is no reported significant difference between batch of <= 5 vs batch of 1, other than 'computational efficiency', and that would directly contradict the need to redo the work in this study.

Further, the evidence reported in the literature is dominantly supportive of using small batches for optimal generalisation performance. Since training time was low for this dataset, we used, what the literature indicated to us, is the optimal batch size.

[1] Wilson, D.R. and Martinez, T.R., 2003. The general inefficiency of batch training for gradient descent learning. *Neural networks*, 16(10), pp.1429-1451.

[2] Keskar, N.S., Mudigere, D., Nocedal, J., Smelyanskiy, M. and Tang, P.T.P., 2016. On large-batch training for deep learning: Generalization gap and sharp minima. *arXiv preprint arXiv:1609.04836*.

[3] Chetlur, S., Woolley, C., Vandermersch, P., Cohen, J., Tran, J., Catanzaro, B. and Shelhamer, E., 2014. cudnn: Efficient primitives for deep learning. *arXiv preprint arXiv:1410.0759*.

4. The weight initialisation method is not clear, please elaborate more.

Thank you for this question. In the manuscript as reviewed (section II. D. paragraph 2), you will find the following statement:

"Linear unit weights were drawn from a real (single precision) uniform distribution in the range $\left[-\sqrt{\frac{3}{fan\ in}} + \sqrt{\frac{3}{fan\ in}}\right]$. ELU unit weights were drawn from a real (single precision) uniform distribution in the range $\left[-\sqrt{\frac{6}{fan\ in}} + \sqrt{\frac{6}{fan\ in}}\right]$, where the *fan in* is the total number of the local (spatial) and feature

inputs to any given unit in a convolutional layer, and the total number of feature inputs to any given unit in a fully connected layer.”

This means that for a linear unit with 100 inputs, we randomly initialised any given weight between $-\sqrt{\frac{3}{100}}$ and $+\sqrt{\frac{3}{100}}$ (where the numerator is 3 for linear and 6 for ELU units). The same applies to a convolutional unit, except we of course include spatial inputs, therefore a 3×3 convolutional unit with 100 feature inputs, we randomly initialised any given weight between $-\sqrt{\frac{6}{100 \times 3 \times 3}}$ and $+\sqrt{\frac{6}{100 \times 3 \times 3}}$ (where again the numerator is 3 for linear and 6 for ELU units).

5. Section F, too technical details that are not helpful

Thank you for the suggestion. We understand the concern that this type of statement, in the modern climate of TensorFlow/Keras, PyTorch, and others, is extremely rare. We respectfully disagree that the technical details of this section are not helpful. This section describes precisely the software and hardware used in our study, correctly attributing credit to the engineers (lead author) and the institution supporting the engineering, Manchester Metropolitan University, rather than to commercial companies like Google (TensorFlow), or Facebook (PyTorch).

, and why not use something like Tensorflow or PyTorch which are standard deep learning software libraries?

Thank you for this question. We agree and are very supportive of the emerging standards for making research accessible via open frameworks for sharing models and code.

We have neither contributed a novel architecture/training method, nor do we intend for the resulting model to be used by us, or others, as it is not appropriate or usable (see discussion). The paper is a demonstration that this problem is solvable using neural network methods, and it does not require the use of TensorFlow or PyTorch.

We have planned a significantly larger study, for which we will use TensorFlow to enable code and model sharing in the public domain.

6. Have the authors considered retraining OpenPose for children pose estimation?

Thank you for this question. Yes, this is a good suggestion, and we have added this to the discussion. The future works we have planned include things like repurposing pre-trained models, and training more sophisticated architectures, some of which is currently in the manuscript as reviewed.

Summary of additional changes to ms.:

- 1) Repositioned figures to accommodate changes to text. Figure 5 is now figure 6 and references to those figures have been updated.
- 2) Removed Ethics section under Methods.
- 3) Added the following sections after Conclusions and before References:
 - a. Ethics Statement
 - b. Data Accessibility

- c. Competing Interests
- d. Authors' Contributions
- e. Acknowledgements
- f. Funding Statement